# Artificial Intelligence in Cardiology—A Narrative Review of Current Status

**DOI:** 10.3390/jcm11133910

**Published:** 2022-07-05

**Authors:** George Koulaouzidis, Tomasz Jadczyk, Dimitris K. Iakovidis, Anastasios Koulaouzidis, Marc Bisnaire, Dafni Charisopoulou

**Affiliations:** 1Department of Biochemical Sciences, Pomeranian Medical University (PMU), 70-204 Szczecin, Poland; koulaou@yahoo.co.uk; 2Division of Cardiology and Structural Heart Diseases, Medical University of Silesia, 40-551 Katowice, Poland; tomasz.jadczyk@gmail.com; 3International Clinical Research Center, St. Anne’s University Hospital Brno, 656 91 Brno, Czech Republic; 4Department of Computer Science and Biomedical Informatics, University of Thessaly, 40500 Lamia, Greece; diakovidis@uth.gr; 5Department of Social Medicine & Public Health, Pomeranian Medical University (PMU), 70-204 Szczecin, Poland; 6Department of Medicine, OUH Svendborg Sygehus, 5700 Svendborg, Denmark; 7Surgical Research Unit, Odense University Hospital, 5000 Odense, Denmark; 8Department of Clinical Research, University of Southern Denmark (SDU), 5000 Odense, Denmark; 9Cardiology Research and Scientific Advancements, UVA Research, Toronto, ON L3R 3Z3, Canada; marc.bisnaire@uvaresearch.com; 10Academic Centre for Congenital Heart Disease, 6500 HB Nijmegen, The Netherlands; dafnithess@yahoo.com; 11Amalia Children’s Hospital, Radboud University Medical Centre, 6525 GA Nijmegen, The Netherlands

**Keywords:** artificial intelligence, cardiology, heart failure, arrythmias, voice technology, cardiac imaging

## Abstract

Artificial intelligence (AI) is an integral part of clinical decision support systems (CDSS), offering methods to approximate human reasoning and computationally infer decisions. Such methods are generally based on medical knowledge, either directly encoded with rules or automatically extracted from medical data using machine learning (ML). ML techniques, such as Artificial Neural Networks (ANNs) and support vector machines (SVMs), are based on mathematical models with parameters that can be optimally tuned using appropriate algorithms. The ever-increasing computational capacity of today’s computer systems enables more complex ML systems with millions of parameters, bringing AI closer to human intelligence. With this objective, the term deep learning (DL) has been introduced to characterize ML based on deep ANN (DNN) architectures with multiple layers of artificial neurons. Despite all of these promises, the impact of AI in current clinical practice is still limited. However, this could change shortly, as the significantly increased papers in AI, machine learning and deep learning in cardiology show. We highlight the significant achievements of recent years in nearly all areas of cardiology and underscore the mounting evidence suggesting how AI will take a central stage in the field.

## 1. Introduction

Artificial intelligence (AI) develops methods rendering computers to engage in human-like thought processes, such as learning and reasoning. Many AI methods are built on machine learning (ML) algorithms which involve processes enabling the AI systems to optimally adapt their parameters in solving a given problem based on training data; in that aspect, they can infer the best possible decision in a given situation. The exponential development of AI, mainly in the subdomains of ML and deep learning (DL), has quickly attracted clinicians’ interest to create new integrated, reliable, and efficient methods for providing quality healthcare. Deep learning is making significant advances in solving problems that have resisted the best attempts of the artificial intelligence community for many years. It has turned out to be very good at discovering intricate structures in high-dimensional data and is, therefore, applicable to many domains in science, business and government [1]. Furthermore, deep learning models can learn from input data, including numbers, text or even combinations of input types [2].

Cardiologists make decisions for patient care from data, and they tend to have access to richer quantitative data on patients compared with many specialties. Despite some potential pitfalls, it is evident that the best way to make decisions based on data is to apply techniques drawn from AI. AI requires close collaboration among computer scientists, clinical investigators, clinicians, other healthcare professionals, and regulatory authorities to identify the most relevant problems to be solved. AI is currently being investigated in several cardiology domains, from clinical decision support systems (CDSS) to imaging interpretation. As a result, researchers have proposed new innovative ideas and practices related to the diagnostic and therapeutic management of cardiovascular diseases, promising ground-breaking developments during the coming years for both cardiovascular sciences and care [3].

Integrating AI into cardiology practice is a change that the profession should embrace. AI has the potential to provide physicians access to actionable data in even greater depth than ever before. Yet, despite the apparent potential, the impact of AI in current clinical practice is still limited. However, this could change shortly given the increased interest of the scientific community in the application of AI in cardiology.

## 2. Terminology

AI is an integral part of many CDSS, providing methods to computationally infer decisions in a way that resembles human inference processes [4]. Such methods are generally based on medical knowledge, either directly encoded with rules or automatically extracted from medical data using ML. ML methods, such as Artificial Neural Networks (ANNs) and Support Vector Machines (SVMs), are based on mathematical models with parameters optimally tuned using the appropriate algorithms. Parameter tuning enables the automatic discovery of rules required for solving a problem, e.g., for predicting a pathological condition. In that aspect, a system can learn from data. ML algorithms can be categorized as supervised if learning is performed on annotated datasets with known, ground truth data that can be used as the gold standard or unsupervised when learning is performed on non-annotated datasets. There are also semi-supervised approaches, where learning is based on annotated and non-annotated data [5].

The ever-increasing computational capacity of today’s computer systems enables more complex ML systems with millions of parameters, bringing AI closer to human intelligence. In this direction, the term DL has been introduced to characterize ML based on deep ANN (DNN) architectures with multiple layers of artificial neurons [6]. Representative DNN architectures, called Convolutional Neural Networks (CNNs), have revolutionized signal analysis by being capable of recognizing meaningful patterns within the signals without being based on subjective handcrafted formulas for feature extraction, e.g., pathological patterns in electrocardiograms (ECG) [7], or in spatiotemporal cardiac imaging [8]. Today’s ML/DL methods used in the context of CDSSs have a high learning capacity; however, their capabilities are limited by the limited availability of high-quality and sufficiently large datasets to validate algorithms, so they perform well for large populations. 

There are two significant obstacles with regard to the availability of datasets: (1) the complexity and usually the inflexibility of the national ethicolegal frameworks for clinical data collection and sharing; and (2) the high effort and costs involved in clinical data annotation. Nevertheless, explainable AI (XAI) can help ensure that patients remain at the center of care and that, together with clinicians, they can make informed and autonomous decisions about their health [9]. The review of Tjoa and Guan reveals an increasing trend towards XAI, with various methods proposed for different medical applications [10]. For example, Maweu et al. recently proposed a framework for explainable classification of ECG patterns [11]. However, it should be noted that the journey towards explainable ML-based CDS is still long, with many challenges ahead. In this direction, promising perspectives arise with the introduction of fuzzy logic in signal interpretation, enabling an uncertainty-aware mapping of linguistic terms to numeric intervals. One successful paradigm combining ML and fuzzy logic for heart disease (HD) detection and heart failure (HF) prediction has been proposed by Vasilakakis et al. [12]. The methodology of that study was extended and applied to develop a generic framework for the explainable classification of medical images that can be applied on any ML-based classification system [13].

## 3. AI in Echocardiography

One of the first applications of AI in echocardiography was the assessment of left ventricular (LV) volume and function with automated quantification. AI may improve imaging quality—and thereby scan and dose time—and assist in segmentation, processing, and analysis. In 2015, Knackstedt et al. used a computer vision vendor-independent software in 255 patients; the software applies a machine-learning algorithm for measuring ejection fraction (EF) and longitudinal strain (LS) from biplane views of the left ventricle [14]. The EF and LS measurements were feasible in 98% of studies, and the average analysis time was 8 ± 1 s/patient, while a 92.1% accuracy was seen compared with the manually traced reference. A year later, Narula et al. investigated the diagnostic value of an ML framework that incorporates speckle-tracking echocardiographic data for automated discrimination of hypertrophic cardiomyopathy (HCM) from physiological hypertrophy seen in athletes [15]. In a cohort of 77 athletes and 62 HCM subjects, the sensitivity and the specificity were 87% and 82%, respectively, for the differential diagnosis between HCM and physiologic hypertrophy. 

Moreover, Zhang et al. used 14,035 echocardiograms performed over ten years to train CNN models for multiple tasks, including automated identification of 23 viewpoints, segmentation of cardiac chambers across five standard views, quantify chamber volumes, LV mass, LV EF and facilitate automatic determination of longitudinal strain through speckle tracking [16]. Automated measurements were comparable (or even superior) to manual ones. Finally, models were developed to detect hypertrophic cardiomyopathy, cardiac amyloidosis, and pulmonary arterial hypertension with C-statistics of 0.93, 0.87, and 0.85, respectively.

Furthermore, some AI models may aid in assessing valvular heart disease. For example, Moghaddasi et al. showed in a cohort of 139 subjects that a Support Vector Machine(SVM) (supervised ML model) classifier had a 99.38% sensitivity and 99.63% specificity to detect severe mitral regurgitation (MR) [17]. Specifically, an accuracy of 99.52%, 99.38%, 99.31%, and 99.59% was recorded to detect normal mitral valve, mild, moderate, and severe MR, respectively. In a proof-of-concept study, Playford et al. used data from 530,871 deidentified echocardiograms derived from 171,571 men and 158,404 women linked to a median of 4.1 years of follow-up to develop an AI system for the assessment of the severity of aortic stenosis (AS) [18]. The AI correctly identified 95.3% of patients with traditional high gradient AS versus 73.9% for the continuity equation. The entire phenotype evaluation obtained the severity of aortic stenosis without reference to LV outflow track velocity or dimension. The algorithm performed equally well in normal and impaired LV systolic function and low-flow and low-gradient severe AS. 

## 4. AI in Cardiac/Coronary Computer Tomography (CCT) 

The coronary artery calcium (CAC) score is a relatively new technique for coronary atherosclerosis identification and risk stratification. The amount of CAC volume can be quantified using the Agatston scoring method when applied to non-contrast ECG-gated coronary CT images [19]. However, image quality can be deteriorated by image noise, motion artifacts, or blooming artifacts from extensively calcified vessels or devices. In addition, the CAC scoring process is often a time-consuming one, making it an ideal candidate for time-saving AI applications. Wolterink et al. proposed an algorithm based on supervised learning to directly identify and quantify coronary artery calcification (CAC) in CCTA without initial coronary artery tree extraction [20]. This study included cardiac CT exams of 250 consecutively scanned patients. The proposed algorithm aimed the automatic CAC quantification in CCTA, reaching a sensitivity of 72%.

Recently, Martin et al. evaluated a novel deep learning-based piece of research software (Automated CaScoring, Siemens Healthineers) for CACS on non-contrast CT images [21]. This approach was based on a convolutional neural network trained on 2000 annotated datasets. As a result, the ML software correctly classified 93.2% of patients (476/511) into the same risk category as the human observers. ML methods were used to assess CACs from non-contrast-enhanced low-dose chest CT performed for lung cancer screening. In a 5973 non-contrast non-ECG gated chest CT scans dataset, a deep CNN was employed to extract the Agatston scores directly from these images [22]. The algorithm yielded a Pearson correlation coefficient of 0.93 and correctly stratified 73% of cases into the corresponding risk category.

CTCA has become the first-line examination in detecting and quantifying coronary stenosis. It has shown excellent sensitivity and negative predictive value for coronary artery stenosis. Currently, the reporting is based on a subjective visual assessment by clinicians.

Van Hamersvelt et al. evaluated the added value of DL analysis of the LV myocardium (LVM) in coronary CT angiography (CCTA) overdetermination of the degree of the coronary stenosis for identification of patients with stress-induced ischemia [23]. Multiple AI techniques were used in this study; firstly, the LV myocardium was automatically segmented using a multiscale CNN. Afterwards, the algorithm unsupervised using a convolutional auto-encoder, LV myocardium, was characterized (encoded) on all CT slices. Finally, patients were classified with a support vector SVM to those with or without functionally significant coronary artery stenosis based on the extracted features. The proposed method improved discrimination (AUC = 0.76) compared to classification based on the determination of the coronary degree of stenosis only (AUC = 0.68). The proposed method’s sensitivity and specificity were 84.6% and 48.4%, respectively. 

In the NXT trial (*Heart Flow analysis of coronary blood flow using CT angiography: NeXt sTeps* trial), 254 subjects underwent CCTA before invasive coronary angiography with fractional flow reserve (FFR), which was the reference standard [24]. The aim was to investigate the associations between coronary stenosis severity, semi-automated assessment of atherosclerotic plaques, derived fractional flow reserve (FFRCT) and lesion-specific ischemia (identified by FFR). An assessment of plaque characteristics was shown to improve the discrimination of lesion-specific ischemia compared with stenosis alone. In a sub-study, the same group investigated if machine learning integration of clinical data, quantitative stenosis and plaque metrics measured from CCT can effectively predict lesion-specific ischemia (identified by FFR) [25]. This combination improved the prediction of lesion-specific ischemia. The integrated ML ischemia risk score exhibited higher AUG (0.84) than individual CTA measures, including stenosis (0.76), low-density noncalcified plaque volume (0.77), total plaque volume (0.74), and pre-test likelihood of coronary artery disease (0.63); *p* < 0.006.

Kelm et al. used an ML algorithm to automatically identify, grade and classify coronary stenosis caused by calcified and non-calcified plaques [26]. Their random forest model was trained on 229 CTA volumes following centerline extraction and lumen segmentation. The model performed accurate stenosis identification and lumen cross-sectional area estimation, with an average processing time of 1.8 s per case. 

Zreik et al. trained a recurrent CNN to detect coronary plaque accurately, determine its composition, and classify the coronary stenosis as obstructive or non-obstructive [27]. For the detection and characterization of coronary plaque, the method has achieved an accuracy of 0.77. For detection of stenosis and determination of its anatomical significance, the method has achieved an accuracy of 0.80.

## 5. AI in Cardiac MRI

After the acquisition of images acquisition, AI can be employed to check that all the acquired images fit a prescribed imaging standard for further processing or analysis. For example, in MRI, investigators have proposed the automatic detection of possible missing basal and apical slices [28], the location of the ascending or descending aorta [29], or the presence of potential interslice motion artefacts and complete heart coverage [30,31]. 

The assessment of EF is a part of cardiac MRI scan analysis, and a trained cardiologist needs up to 20 min to complete it. Therefore, a fully automated approach will greatly value interpreting images rapidly, preventing interobserver and intraobserver interpretation variance. 

Xue et al. proposed a robust method for LV detection using the CNN [32]. The CNN models were trained by assembling 25,027 scans. The model with three classes (3CS) for background, LV, and RV, successfully detected the LV in 99.98% of all test cases.

Tan et al. successfully developed a fully automated neural network regression-based algorithm for segmentation of the LV in cardiac MRI, with full coverage from apex to base across all cardiac phases, utilizing both short-axis (SA) and long-axis (LA) scans [33]. First, a network was trained to automatically identify the myocardium and detect the centre of the cavity. Then another network was trained to estimate radii from the cavity centre, producing smooth epicardial and endocardial contours. A similar approach was proposed by Du et al., where a boundary regression was performed on both left and right ventricles on short-axis images producing contours instead of pixel classification [34].

Furthermore, Bernard et al. published an automated segmentation algorithm, achieving Dice similarity coefficients of 0.95, or better, when compared to manual tracing [35]. Fahmy et al. presented an algorithm based on deep convolutional neural (DCN) networks to automatically quantify LV mass and scar volume on late gadolinium enhancement (LGE) in patients with HCM [36]. 

Recently, increased interest has been noted in the novel field of *radiomics*. The term *radiomics* reflects a process of converting digital medical images into mineable high-dimensional data by extracting a high number of handcrafted quantitative imaging features based on a wide range of mathematical and statistical methods [37]. Radiomics can be an efficient tool for discriminating between hypertensive heart disease and HCM patients when radiomics is applied on T1 and T2 mapping [38]. 

## 6. AI in Nuclear Cardiac Imaging

Myocardial perfusion imaging (MPI) by single-photon emission computed tomography (SPECT) and positron emission tomography (PET) plays a crucial role in the diagnosis and management of CAD. In nuclear cardiology, AI models demonstrated the use of ML to improve diagnostic accuracy, identify perfusion defects and their location, and predict early revascularization. In nuclear cardiac imaging, an ML algorithm (LogitBoost) was proposed to predict those patients with suspected coronary artery disease in whom early revascularization can be effective [39]. 713 single-photon emission computed tomography (SPECT) myocardial perfusion imaging (MPI) correlating invasive angiography with 372 revascularization events was used. The prediction of revascularization by the ML algorithm was compared to the analysis by two experienced readers utilizing all imaging, quantitative, and clinical data. The AUC for the ML was similar to one reader and superior to the second reader.

DL was used to predict obstructive disease from SPECT MPI [40] automatically. A total of 1638 patients without known CAD, undergoing SPECT MPI and invasive coronary angiography were studied. AUC for disease prediction by DL was significantly higher than for the total perfusion deficit (0.80 vs. 0.78 per person and 0.76 vs. 0.73 per vessel). In addition, the time needed to evaluate a new patient with this model was <1 s.

Another study aimed to determine whether ANN might help to diagnose coronary artery disease [41]. The ANN was trained to classify potentially abnormal areas as true or false based on the nuclear cardiology expert interpretation of 1001 gated SPECT images. The diagnostic accuracy of the ANN was compared with 364 expert interpretations that served as the gold standard of abnormality for the validation study. The ANN identified stress defects better than nuclear cardiology experts (AUC = 0.92 vs. 0.82, respectively).

## 7. AI in Heart Failure

Sanchez-Martinez et al. showed that an unsupervised ML algorithm for analyzing left ventricular (LV) long-axis function at rest and exercise could facilitate the diagnosis of heart failure (HF) with preserved ejection fraction (HFpEF) [42]. Inan et al. showed that the telemonitoring of HF patients in combination with ML algorithms can assess compensated and decompensated HF states [43]. A Swedish study confirmed that ML algorithms could identify four phenotypes with different clinical courses and therapeutic responses [44].

Koulaouzidis et al. developed an algorithm for detecting patients at high risk of HF hospitalization, using daily collected physiological data (blood pressure, heart rate, weight) by non-invasive telemedicine [45]. The algorithm could identify such patterns and classify them as abnormal by assessing the predictive value of each of the monitored signals and their combinations using analysis of vectors (e.g., vectors of raw signal values, vectors of signals obtained by Multi-Resolution Analysis). The best predictive results were achieved with the combined used of weight and diastolic BP. The highest predictive performance was achieved using eight-day TM data (area under the receiver operator characteristic curve (AUC) 0.82 ± 0.02).

Finally, the clinical response to cardiac resynchronization can be predicted using ML systems [46].

## 8. AI in Arrythmias 

One of the most widespread applications of AI in cardiology is the prediction of cardiac arrhythmias. An effective ML-based methodology to classify ECG was proposed to predict the onset of paroxysmal atrial fibrillation [47]. The prediction performance of this algorithm was superior to the previously developed methods in terms of both the sensitivity and specificity, which was 100% and 95.5%, respectively. DL techniques have succeeded in detecting cardiac arrhythmia (17 classes) based on long-duration ECG signal analysis [48].

Attia et al. from the Mayo Clinic used paired 12-lead ECG and echocardiogram data, including the left ventricular ejection fraction (LVEF) from 44,959 patients, to train a CNN to identify patients with ventricular dysfunction, defined as LVΕF ≤ 35%, using the ECG data alone [49]. When tested on an independent set of 52,870 patients, the network model yielded values for AUC, sensitivity, and specificity of 0.93, 86.3%, and 85.7%, respectively. In another study from the Mayo Clinic, a CNN was trained and validated using a digital 12-lead ECG from 2448 patients with a verified HCM diagnosis and 51,153 non-HCM age- and sex-matched control subjects [50]. The algorithm detected adult HCM patients from the ECG with an AUC of 0.96, a sensitivity of 87% and a specificity of 90%. The performance of the above-described algorithm was tested in a cohort of 300 children and >18,000 age- and sex-matched controls [51]. The results were impressive: the AUC of the model for HCM detection was 0.98, with 92% sensitivity and 95% specificity; positive predictive value (PPV) was 22%, and negative predictive value (NPV) was 99%. It tended to work better in adolescents than in small children.

Finally, Koulaouzidis et al. developed a statistical index based on the phase space reconstruction (PSR) of the ECG for the accurate and timely diagnosis of ventricular tachycardia (VT) and ventricular fibrillation (VF) [52].

Recently, a breakthrough technology, Cardio-HART™, was introduced. Using bio-signals enhanced with AI, it is possible to estimate echocardiographic-like findings (HART-findings), including structural, functional and hemodynamical abnormalities. Functional abnormalities mean systolic and diastolic dysfunction supported by measurements of LVEF, GLS, E/e’, LAVI and wall motion abnormalities. The detection of structural and functional cardiac abnormalities with promising performance supports the early detection of HF. Significantly, however, being bio-signal based, it can be used in primary care settings, since ECHO is only available in Secondary Care.

Cardio-HART™ is based on complementing ECG with phonocardiograph (PCG) and a novel bio-signal of a physiological nature (MCG), shows significantly higher sensitivity in detection of common heart diseases. AI make fusion of each bio-signals own strengths:

ECG—Electro-physiological abnormalities: arrhythmias, premature beats, atrioventricular blocks, bundle-branch blocks, etc.;

PCG—Hemodynamical diseases: valve stenosis, valve regurgitation, hypertension (arterial or pulmonary hypertension);

MCG—Mechano-physiological abnormalities: cardiomyopathy, myocarditis, myocardial infarction, ischemia, hypertrophies, atrial enlargements, systolic or diastolic dysfunction, or other wall motion problems;

The performance is summarized by using triple classification of the echocardiographic summary, where the middle category between Normal and Abnormal, called “Mild”, means that the patient has one or a few mild abnormalities, typically diastolic dysfunction with preserved LVEF and/or some asymptomatic heart enlargement or non-significant valve insufficiency (Table 1 and Table 2). The ECG summary with its triple outcome is provided in statistics for comparison purposes: normal, borderline, and abnormal. 

## 9. Voice Technology in Clinical Practice

Synthesis of ML and computational linguistics enabled human-computer voice interaction. Speech recognition algorithms interpret the complexity of conversational language while text-to-speech technology leverages neural network techniques to generate human-like voices. This is crystalized in the form of voice assistants (VAs) exemplified by Amazon Alexa or Google Assistant; these advanced software architectures are available on smart speakers (Amazon Echo, Google Home, respectively) or smartphones [53]. Broad market adoption and strong consumer interest in using smart speakers for daily tasks created an opportunity to implement voice technology in health care. VAs, also known as conversational agents or voice bots, are emerging tools for the remote delivery of medical services already adopted into clinical practice.

The FDA classifies medical software based on the impact on the patient, illustrating the application’s main functionality [54]. A wide spectrum of AI-driven voice applications ranges from educational services through optimization of processes and data collection to the Software as a Medical Device (SaMD) solutions. Figure 1 illustrates a risk-based approach for the evaluation and categorization of medical software. Notably, the software transferring medical data must comply with the Health Insurance Portability and Accountability Act of 1996 (HIPAA) [55] and the General Data Protection Regulation (GDPR) [56] in the USA and Europe, respectively.

Educational voice applications can be exemplified by services providing instruction or answering frequently asked questions. For example, the Mayo Clinic First Aid skill provides guidelines on cardio-pulmonary resuscitation procedures [57]. Furthermore, a clinically validated tool developed by the Mayo Clinic and the Centers for Disease Control and Prevention (CDC) experts advises on COVID-19 where patients can verbally ask questions related to the SARS-CoV2 virus [58]. Furthermore, the voice-enabled technology is used to streamline repetitive tasks helping medical professionals reduce the administrative burden within their practices. For example, the Giant Eagle Pharmacy skill for Amazon Alexa allows patients to set up medication reminders and refill prescriptions [59].

Moreover, to improve organization workflow, Providence St. Joseph Health makes an appointment scheduling system available on Amazon Alexa-enabled devices, advancing convenient access to health care [60]. Additionally, to optimize in-hospital logistics, the voice-powered Aiva Health [61], OrbitaASSIST [61], and Vocera Engage [62] skills serve as digital bedside assistants that keep patients connected to the clinical team. Accordingly, the hospitalized individuals and caregivers can speak requests for a care team member supporting effective routing.

VAs have also been shown to support paperless medical history documentation at the Cardiology Outpatient Clinic of the Cedars-Sinai Medical Center in Los Angeles (CA, USA). The HIPAA-compliant CardioCube voice application deployed on Amazon Echo was integrated with electronic health records (EHR) to automate the admission process, during which patients verbally answered a set of pre-specified clinical questions. Study results showed that CardioCube collected medical information obtained during human-VA conversation with a high accuracy generating ready-to-use reports in the EHR system [63]. 

Importantly, the voice AI solutions can be integrated with an existing health care ecosystem. For example, the FCNcare application by CardioCube reviewed by the FDA was implemented at the Family Care Network in Bellingham (WA, USA), enabling heart failure and diabetic patients to update medical status and biometric data (blood pressure, glucose level) from home through a HIPAA-compliant conversational agent integrated with a CDSS and EHR system. Focused medical reports were instantly available for review by a telemedical nurse who was taking adequate clinical actions based on received information. In addition, automatic analysis of the patient-reported outcomes and red-flagging system provided a routinely used digital screening tool for a long-term follow-up of chronic patients [64]. 

## 10. AI and Non-Cardiac Conditions

Dyselectrolytaemia represents a broad spectrum of electrolyte disbalance associated with high cardiovascular risk [65,66]. Abnormal potassium level (dyskalaemia) is one of the most common findings among patients presenting to emergency departments [67,68]. ECG manifestation of hypokalemia includes widespread ST depression, increased P wave amplitude, and T wave inversion. At the same time, hyperkalemia promotes increased T wave amplitude and P wave and QRS complex widening [69]. Despite well- described morphologic features, diagnosing dyskalaemias may be challenging, especially since only half of the hypokalemic patients present evident ECG abnormalities [70]. Interestingly, an AI-based system designed to detect dyskalaemia on ECG identifies moderate-to-severe hypo- and hyperkalemia [71]. Moreover, a deep learning model showed high performance in detecting ECG manifestation of hyper- and hyponatremia and hyper- and hypocalcemia [72]. In addition to electrolyte imbalance, the ML approach was proven to detect overt hyperthyroidism [73], which can open access to non-invasive screening, early diagnosis and treatment to prevent serious cardiovascular complications (i.e., cardiac arrhythmia, heart failure, and stroke) [74,75]. Furthermore, the AI algorithm can mark pathophysiological conditions beyond dyselectrolytaemia as an independent predictor of adverse outcomes [71].

## 11. Discussion

AI methods will be used in two main areas in cardiology: automating tasks that humans might otherwise perform and generating clinically crucial new knowledge for risk prediction. In addition, the AI technology will assist cardiologists in making prompt diagnoses, improving interpretation skills, and delivering more accurate and personalized care. Major ethical issues could arise from the implication of AI predictive models in clinical practice. Although researchers continue to improve the accuracy of clinical predictions, a perfectly calibrated prediction model may not significantly enhance clinical care. An accurate prediction of a patient outcome does not tell us what to do if we want to change that outcome. We cannot even assume whether possible changes will affect the predicted outcomes.

After an algorithm/model is built, the next important step is determining whether the model remains accurate if new data are fed into the model. For this reason, data are needed for both training (developing the model) and for testing (assessing how well the model continues to predict the same outcome with the new data). It is essential that the data that was used for training is not used for testing. There is a risk of overfitting; the model may only be suited to the dataset from which it was developed. The development of AI predictive models is based on data from primarily white patients, and these data may be less precise in minorities. Most evidence-based treatment recommendations for HF are based on studies on White men. An AI algorithm based on preexisting biases will increase uncertainty, primarily when used for complex conditions. Another significant ethical issue is the question of accountability. When an AI system fails at a particular assigned task, and this failure has substantial clinical implications, who should be responsible? The manufacturer who built it, the data owner, or the clinician who used it? Who is accountable when things go wrong with AI remains an unanswered question.

The acceptance of AI by the patient needs to be explored in depth before adopting AI predictive models in clinical practice. It is well known that good doctor-patient relationships positively affect health outcomes. It will be difficult for the patients to understand why an AI system suggests a particular diagnosis, treatment, or outcome prediction. But, more importantly, it will be difficult for the patients to accept that an AI system excluded them from treatment. This will have a negative impact on the confidence of the patients in the medical society. Even if AI is successfully introduced in clinical practice, human interaction will still be required in healthcare, particularly in cases with social interaction and a demand for a holistic perspective. Until now, AI systems have not considered patient fears and worries, the social environment, lifestyle and other factors. It remains to be determined how and to what extent AI can address these factors.

## 12. Conclusions

AI techniques are of great interest in the healthcare and medical field because they can use sophisticated algorithms to analyse large volumes of physiological data obtained from thousands of patients, thus gathering information to assist clinical practice and decision-making. Considering that the performance of AI technology depends on the appropriate data being used and their quality, it is important to focus on solutions for more flexible and larger-scale data sharing and methods with less requirements for human annotation, such as semi-supervised and transfer learning methods. Finally, although AI may predict likely outcomes, the automatic generation of management decisions does not seem desirable, as personal, environmental and social aspects of each patient differ significantly.

## Figures and Tables

**Figure 1 jcm-11-03910-f001:**
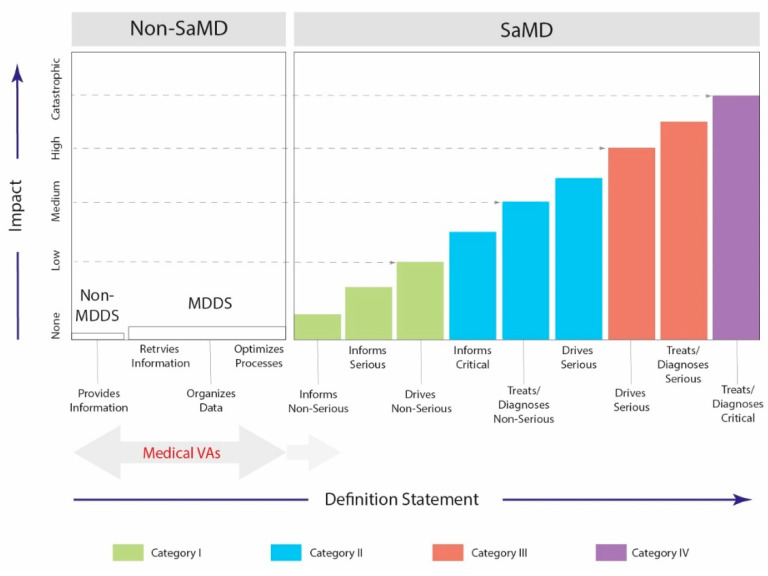
Risk-based approach to evaluate and categorize software designed for medical purposes. Category I-IV reflects an impact on patients being associated with medical utilization (inform, drive, treat, diagnose) and clinical scenarios that the service is intended for (non-serious, serious, critical). Non-MDDS—Non-Medical Device Data Systems; MDDS—Medical Device Data Systems; SaMD—Software as a Medical Device.

**Table 1 jcm-11-03910-t001:** Confusion matrix of Triple Aggregated Diagnostic of HART Summary and ECG.

		**HART Summary**			
		**Negative**	**Positive/Mild**	**Positive/Abnormal**	**Sum**
**ECHO ground truth**	Normal	1916	974	60	2950
	Mild	632	1426	386	3444
	Abnormal	68	1288	2010	3366
					9760
		**ECG summary**			
		**Normal**	**Borderline**	**Abnormal**	**Sum**
**ECHO ground truth**	Normal	1654	670	626	2950
	Mild	1282	1074	1088	3444
	Abnormal	426	534	2406	3366
					9760

**Summary**. (i) The negative HART summary has high negative predictive value for the abnormal patient, while the positive/abnormal HART summary has high positive predictive value for the non-normal patients. (ii) When the HART assessment is negative (normal), then the moderate/severe abnormality can be ruled out, as the patient would be normal by echocardiography with high probability. The discussion can be only between normal and mild. (iii) When the HART assessment is positive/abnormal, then the patient has some moderate/severe condition with high probability and this indicates that the patient needs to be referred to cardiology for a detailed and more certain diagnosis. (iv) When the HART assessment is positive /mild, then the patient’s condition is less certain, and it is advised to carefully consider the ECG, PCG and MCG findings together with patient symptoms and history for the appropriate referral and treatment options.

**Table 2 jcm-11-03910-t002:** Probability evaluation of HART and ECG summary compared to ECHO-based ground truth. CHART: Cardio-Hart ECG: electrocardiogram, ECHO: echocardiogram, NPV: negative predictive value, PPV: positive predictive value.

Outcome	by HART Summary	by ECG Summary
Negative	Patient predicted as negative by HART is not abnormal in 97.4% probability (NPV = 97.4%)	Patient predicted as normal by ECG is not abnormal with 87.3% probability (NPV = 87.3%)
	Patient predicted as negative by HART is mild with 24.1% probability	Patient predicted as normal by ECG is mild with 38.1% probability
	Patient predicted as negative by HART is abnormal only with 2.6% probability (100%-NPV)	Patient predicted as normal by ECG is abnormal only with 12.7% probability (100%-NPV)
Mild	Patient predicted as positive/mild by HART is mild with 52% probability	Patient predicted as borderline by ECG is mild with 47% probability
Positive	Patient predicted as positive/abnormal by HART is not Normal with 97.6% probability (PPV = 97.6%)	Patient predicted as abnormal by ECG is not normal with 84.8% probability (PPV = 84.8%)
	Patient predicted as positive/abnormal by HART is mild with 15.7% probability	Patient predicted as abnormal by ECG is mild with 26.4% probability
	Patient predicted as positive/abnormal by HART is normal only with 2.4% probability (100%-PPV)	Patient predicted as abnormal by ECG is normal with 15.2% probability (100%-PPV)

## Data Availability

Not applicable.

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
