# Peer review of "Artificial Intelligence in Cardiology—A Narrative Review of Current Status"

_jcm, 2022, doi:10.3390/jcm11133910_

Round 1

Reviewer 1 Report

Major comments

This review article summarizes the significant achievements of recent years in many fields of cardiology including echocardiogram, MRI, nuclear cardiology, arrhythmia, heart failure, and voice technology. It would be better if they can simplify this article by listing only representative studies and shortening each section. Instead of listing studies, what kind of task can be done by AI and what is the main issue in AI analysis in each field can be discussed.  The section to discuss future clinical implication of AI and pitfall/obstacle of AI can be added.

Minor comments

In introduction, authors mentioned that "the impact if AI in current clinical practice is limited". The reason for this should be explained.

In conclusions, authors mentioned that "The AI technology should not be seen as a replacement for cardiac specialist". The reason for this should be explained.

SPRCT is probably a typo for SPECT.

Author Response

your suggestions have been taken into account and an enhanced version of the manuscript has been submitted.

Reviewer 2 Report

This topic is important and I want to encourage author contributions for writing this paper. However, the article is needed a substantial revision in article structure. My specific comments are mentioned below.

1. Introduction: the author should describe the most significant change of this deep learning revolution. For example, you can refer these paper.

Nature volume 521, pages436–444 (2015)

Nature Medicine volume 28, pages31–38 (2022)

2. Introduction: the author should point out the most important AI application in cardiology, For example, you can refer this paper.

European Heart Journal, Volume 43, Issue 4, 21 January 2022, Pages 271–279

3. From the section 3 to section 9, author introduced the AI in many fields but they are difficult to follow the logic. For example, sections 7 and 8 described diseases and sections 3 to 6 described examinations. I suggest to use following subtile in this manuscript as following based on the complexity. ECG, chest X-ray, cardioechogram, CCT, CMRI, Nuclear cardiac imaging, etc.

4. For each examination, I suggest to use a table to compare all important papers and try to mention the contributions from major research teams (published more than 3 papers in one AI applications). For example in AI-enabled ECG system, Mayo clinic obviously make the greatest contribution, and I know of at least 3 teams with more than 10 relevant publications. Following link can be used to find these papers.

https://scholar.google.com.tw/citations?user=7iRgI1IAAAAJ

https://scholar.google.com.tw/citations?user=DMd-2NEAAAAJ

It's important to note that to directly use publications from above researchers doesn't mean that's enough, try doing some more reviews, especailly in other examinations.

Author Response

your suggestions for this narrative review have been taken into account and an enhanced version of the manuscript has been submitted.

Reviewer 3 Report

Very interesting discussion about AI and its implementation in clinic and in practice.

One of the topics that was not included in discussion is about accuracy of predictions? How many patient-specific data are required for accurate predictions? Are other data e.g. patients medical history included ? How high is the possibility of a patient to be treated as an "exception" in AI based clinical decision making?

Author Response

Thank you for the insightful comment and suggestions

We have taken the chance to revise our manuscript, to include a couple of extended paragraphs in the discussion section (in the tracked changes version highlighted in red), for the use of voice in

Furthermore, in response to your comment/criticism: 'How high is the possibility of a patient to be treated as an "exception" in AI based clinical decision making?' we added the following paragraph:

One central theme to be addressed is how to balance the benefits and risks of AI technology in prediction and decision making. Even if AI models are proven effective and validated, practical issues exist in their clinical deployment. Although there is an opportunity to improve the efficiency and quality of patient care, there is also a risk of discrimination and malpractice. Patients may be excluded from medical treatments based on the prediction models.

Furthermore, AI health predictions can also lead to psychological harm. For example, many people could be traumatized and change behaviour if they learn that they will likely suffer significant health problems. If AI generates predictions about people's health, information may be included in electronic health records one day. Anyone with access to health records could then see predictions about the disease's outcome or predictions for future diseases. Several clinicians and administrators see patients' medical records in the course of medical treatment.

Additionally, patients themselves often authorize others to access their records, such as applying for employment or life insurance. Employers are interested in workers who will be healthy and productive, with few absences and low medical costs. If they believe certain applicants will develop diseases in the future; they will likely reject them. Lenders, landlords, life insurers and others might likewise make adverse decisions about individuals based on AI predictions.

AI technology must be used as a supportive tool and not as a replacement for a physician.

I hope you find these changes satisfactory.

With kind regards

Round 2

Reviewer 1 Report

The presented manuscript is revised adequately.

Author Response

We would only like to thank the reviewer for this final comment. Appreciate your time, indeed, with our manuscript.

Reviewer 2 Report

Authors do not response all comment in current version. For example, I suggested to use following subtitle in this manuscript as following based on the complexity. ECG, chest X-ray, cardioechogram, CCT, CMRI, Nuclear cardiac imaging, etc. However, authors do not follow this suggestion and provide suitable reason. 

Author Response

Dear Reviewer,
Thank you for pointing out that we have not followed the breath of your recommendation(s) from round 1. 
In fact, this was because, as clearly stated in the title, this manuscript aims to give an overview of the status in a fast-pacing field. Furthermore, being narrative has the inherent limitation of bias in presenting only part (those considered more relevant). Therefore, although crucial for the sake of a more systematic revision manuscript of the field, we think that your recommendations will not significantly alter the readability or citability of this narrative manuscript; hence, we elected to stick with our initial presentation. We hope you will be able to accept this clarification.

with kind regards